# Quinoid Pigments of Sea Urchins *Scaphechinus mirabilis* and *Strongylocentrotus intermedius*: Biological Activity and Potential Applications

**DOI:** 10.3390/md20100611

**Published:** 2022-09-28

**Authors:** Natalya V. Ageenko, Konstantin V. Kiselev, Nelly A. Odintsova

**Affiliations:** 1Laboratory of Cytotechnology, National Scientific Center of Marine Biology, Federal State Budgetary Institution of Science, The Far Eastern Branch of the Russian Academy of Sciences (FEB RAS), 690041 Vladivostok, Russia; 2Laboratory of Biotechnology, Federal Scientific Center of the East Asia Terrestrial Biodiversity, Federal State Budgetary Institution of Science, FEB RAS, 690022 Vladivostok, Russia

**Keywords:** biological active substances (BAS), echinoids, naphthoquinones, sea urchins, pigment cells

## Abstract

This review presents literature data: the history of the discovery of quinoid compounds, their biosynthesis and biological activity. Special attention is paid to the description of the quinoid pigments of the sea urchins *Scaphechinus mirabilis* (from the family *Scutellidae*) and *Strongylocentrotus intermedius* (from the family *Strongylocentrotidae*). The marine environment is considered one of the most important sources of natural bioactive compounds with extremely rich biodiversity. Primary- and some secondary-mouthed animals contain very high concentrations of new biologically active substances, many of which are of significant potential interest for medical purposes. The quinone pigments are products of the secondary metabolism of marine animals, can have complex structures and become the basis for the development of new natural products in echinoids that are modulators of chemical interactions and possible active ingredients in medicinal preparations. More than 5000 chemical compounds with high pharmacological potential have been isolated and described from marine organisms. There are three well known ways of naphthoquinone biosynthesis—polyketide, shikimate and mevalonate. The polyketide pathway is the biosynthesis pathway of various quinones. The shikimate pathway is the main pathway in the biosynthesis of naphthoquinones. It should be noted that all quinoid compounds in plants and animals can be synthesized by various ways of biosynthesis.

## 1. Introduction

Quinoid compounds are an important class of organic compounds that have attracted the attention of researchers for many years [1,2]. This is due to the possibility of their practical application as biologically active and medicinal substances, stabilizers in the polymer industry, dyes, reagents in organic synthesis, dehydrating agents and complexing agents. Due to the peculiarities of the living conditions, often due to the high-pressure habitat and specific food preferences, marine aquatic organisms are a new source of chemical compounds, potential medicines and cosmetics, biologically active substances (BAS) and functional foods. These compounds are theoretically interesting for their high chemical activity, the ability to form complexes with charge transfer, etc.

Of particular interest are heterocyclic quinoid compounds and 1,4-naphthoquinone derivatives. This is due to the fact that many natural- and synthetic-condensed quinones have potential biological activity and there is a possibility of studying their redox and complexing properties. In addition, these structures are part of a number of antibiotics and alkaloids of marine organisms, and they are also successfully used in modern medicine and technology. For example, 1,4-naphthoquinone derivatives can be inhibitors of the phosphatase CDC25B [3], which is involved in the regulation of the cell cycle and in oncogenesis [4]. Based on computer modeling methods, a possible binding site of compounds of this class (1,4-naphthoquinone derivatives) with CDC25B protein was indicated [5].

Sources of such valuable BAS are the inhabitants of massive Pacific species—sponges, mollusks and echinoderms. Secondary metabolites isolated from them often perform protective functions against threats, such as predator attacks, biological fouling, microbial infections. Of particular interest to us are sea urchins—*Scaphechinus mirabilis* (*Agassiz*, 1863) and *Strongylocentrotus intermedius* (*Agassiz*, 1863)—because they have quinone pigments (echinochrome A and different spinochromes).

## 2. Results

### 2.1. Sea Urchins—S. mirabilis and S. intermedius

Sea urchins (*Echinoidea*) appeared on Earth about 500 million years ago. They are divided into two types: right and wrong sea urchins.

#### 2.1.1. Flat Sea Urchin—*S. mirabilis*

The first flat sea urchins (sand dollar) appeared about 30 million years ago in West Africa.

Then they spread all over the coast. The irregular (wrong) sea urchin *S. mirabilis* (*Scutellidae*) (Figure 1) is one of the widespread representatives of shallow-water benthos [6]. *S. mirabilis* lives in the Sea of Japan and forms stable settlements on the coast of the southwestern part of Peter the Great Bay at depths from 0.5 to 125.0 m, but depths of 3–6 m are preferred. The thermophilic species of this sand dollar lives exclusively on the surface layer of sandy soil, avoiding silty soil. Sand dollars live in surf zones with sharp fluctuations in salinity. In fact, the sandy *S. mirabilis* is located on the surface of the bottom, but sometimes it burrows into the sand to a depth of 1–4 cm.

In addition, these animals are exposed to changing environmental factors during their life cycle. They can be exposed to seawater with significantly reduced salinity at various depths in hard soils. The size of the flat shell of adult *S. mirabilis* reaches 50–70 mm in diameter and up to 1 cm in thickness [6]. The thick shell of this sea urchin is covered with small and dense needles of dark purple color. Its diet consists of algae and detritus [7]. Spawning of this type of sand dollar occurs in the warm summer period from mid-July to the end of August in the Sea of Japan. After the fertilization of the *S. mirabilis* egg, a symmetrical pluteus larva formed on the third day. These larvae are carried by the current or move independently thanks to the cilia. After a few weeks, the larvae sink to the sandy bottom and turn into small round animals.

#### 2.1.2. Gray Sea Urchin—*S. intermedius*

The regular (right) gray sea urchin *S. intermedius* (*Strongylocentrotidae*) (Figure 2) is common in shallow coastal zones of the southern part of the Sea of Okhotsk and the Sea of Japan.

This species of sea urchins mainly lives on rocky areas of the bottom, but sometimes they live on a sandy surface and in thickets of seagrasses [8,9,10,11,12,13,14,15]. The body of this sea urchin has a regular spherical shape, slightly flattened from the side of the mouth opening. The diameter of an adult sexually mature individual is from 3 to 8 cm, and the mass reaches 160–170 g. The color of the needles and the shell of *S. intermedius* is very diverse—red, purple, green, gray, brown. At a young age, these sea urchins feed on films of microscopic algae, and the adults feed on brown algae [8,9,10,11,12,13,14,15,16].

The spawning of this species of sea urchins occurs at different times, depending on the habitat. For example, in the Sea of Japan, the reproductive season is observed in May–June and September–October, and in the Sea of Okhotsk it lasts from June to October. After the fertilization of gray sea urchin eggs, the resulting embryo turns into a pluteus larva after 48 h. Further, from the end of July to the beginning of September, the larvae begin to settle. The process of their settlement ends by November. When settling on any substrate, the larva acquires external radial symmetry and new organs. Further, the larvae grow slowly, reaching a mass of 0.16 g and an average shell diameter of 0.65 cm by one year of life. In the period from July to September, gray sea urchins grow more intensively than in winter. At the age of three, sea urchins become sexually mature, and the size of the shell increases [8].

### 2.2. Naphthoquinoid Pigments of Sea Urchins

Sea urchins, in particular *S. mirabilis* and *S. intermedius*, contain polyhydroxylated naphthoquinoid pigments, which are specific metabolites for this class of echinoderms [17,18]. These pigments are present in the soft and skeletal areas of sea urchins. In addition, naphthoquinones have also been found in starfish. The main ones are echinochrome A (**1**) and five spinochromes A (**2**), B (**3**), C (**4**), D (**5**) and E (**6**) (Table 1 and Table 2).

Naphthoquinones of sea urchins differ from naphthoquinones of other marine animals by the presence in the structure of a large number of free hydroxyl groups and high antioxidant properties [19,20].

In 1885, Macman first discovered a red naphthoquinoid pigment in the coelomic fluid of the sea urchin *Echinus esculentus* (*Linnaeus*, 1758) [21]. German scientists Kuhn and Wallenfels established the structure of this pigment as 2,3,6-trihydroxy-7-ethyl-1,4-naphthoquinone, but later its structure was precisely determined as 2,3,5,6,8-pentahydroxy-7-ethyl-1,4-naphthoquinone—echinochrome A (**1**) (Table 1) [22].

Then a separate group of other naphthoquinoid pigments—spinochromes—were isolated from the ovules, internal organs, shells and needles of various species of sea urchins (Table 1 and Table 2) [1,23,24,25]. Later, the only naphthazarin was isolated from the sea urchins *Echinothrix diadema* (*Linnaeus*, 1758) and *E. calamaris* (*Pallas*, 1774)—spinochrome—with a pyran cycle with a naphthazarin fragment (2-hydroxy-2′-methyl-*2′H*-pyrano[2,3-*b*] naphthazarin) [26]. In addition, methoxy derivatives of spinochromes were found in the sea urchins *Diadema setosum* (*Leske*, 1778) and *D. savignyi* (*Audouin,* 1829), which were previously isolated from starfish, holothurians and ophiurians [27]. The first two of monomethyl ethers of echinochrome A were isolated from the sea urchin *D*. *antillarum* (*Philippi*, 1845) [28]. The following naphthoquinoid pigments were isolated from sea urchins—*Spatangus purpureus* (*Muller*, 1776), *Strongylocentrotus droebachiensis* (*Muller*, 1776) and *S. intermedius*. On the basis of spectral analysis, the structures of the isolated pigments were determined as dimeric spinochromes (binaphthoquinones) [29,30]—ethylidene-3,3′-bis(2,6,7-trihydroxynaphthazarin), its anhydro derivative [28], ethylidene-6,6′-bis(2,3,7-trihydroxynaphthazarin) and 7,7′-anhydroethylidene-6,6′- bis (2,3,7-trihydroxynaphthazarin) [31,32]. A diverse set of naphthoquinones of sea urchins was later supplemented—echinamines A and B (spinochromes with a primary amino group) [33], spinazarin and ethylspinazarin [34] from the flat sea urchin *S. mirabilis*; aminopentahydroxynaphthoquinone and aminoacetyltrihydroxynaphthoquinone (aminated spinochromes) from the sea urchin *Strongylocentrotus nudus* (*Agassiz*, 1864) [35]; sulfated derivatives of spinochromes B and E [23] from the sea urchin *Psammechinus miliaris* (*Muller*, 1771).

The authors of the article characterized the quinoid pigment compounds isolated from the sea urchins *S. intermedius* and *S. mirabilis* of the Sea of Japan [36].

The total extract of the gray sea urchin *S. intermedius* contained spinochromes A–E (**2**–**6**), binaphtoquinones (**7**–**9**) (up to 40%) (Figure 3) and an unknown pigment (**10**). The same qualitative composition of naphthoquinones was isolated from another species of sea urchins *Mesocentrotus nudus* (*Agassiz*, 1863), the extract of which additionally contained other quinoid pigments—echinochrome A (**1**), spinamin E (**11**) (Figure 3) [36].

Binaphthoquinones (**7**–**9**) were also isolated from the sea urchin of *S. mirabilis* by various solvents, as in the sea urchins *S. intermedius*. In addition, extracts of these flat sea urchins contained spinochrome D (**5**) and echinamines A (**12**) and B (**13**) (Figure 3). However, when *S. mirabilis* chloroform is extracted, the main naphthoquinoid pigments are echinochrome A. An almost similar qualitative composition of quinoid compounds was isolated from another flat sea urchin *S. griseus* (*Mortensen*, 1927), belonging to the species *S. mirabilis*. Common extracts from the sea urchins *S. griseus* also contained echinochrome A (**1**), spinochromes D (**5**) and E (**6**), and binaphthoquinones (**7**–**9**) [36].

**Figure 3 marinedrugs-20-00611-f003:**
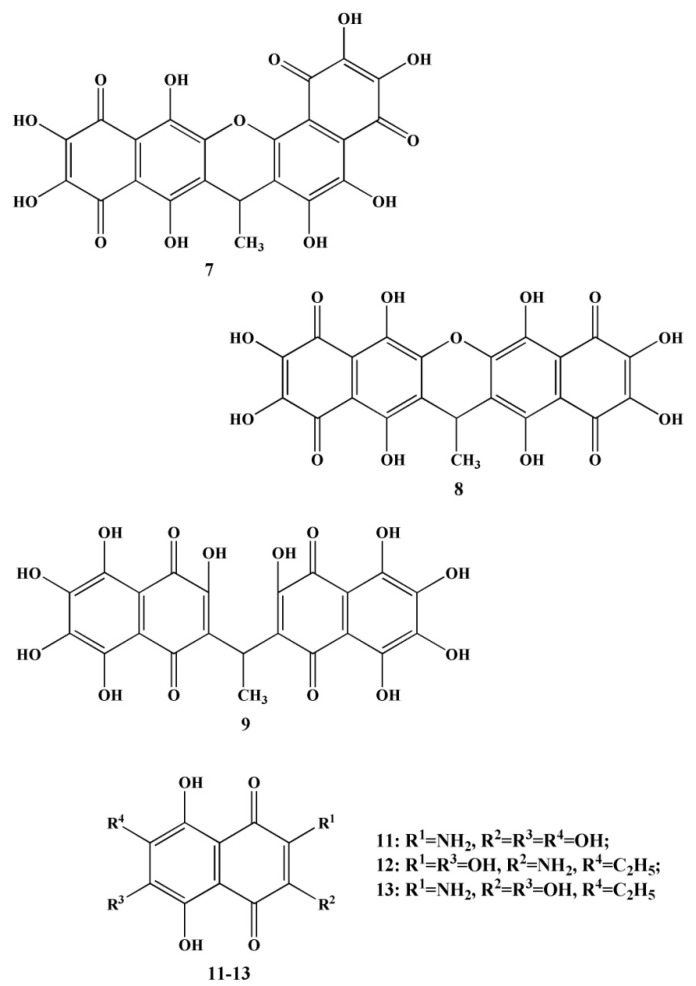
Structures of quinonoid pigments (binaphthoquinones—**7**–**9**, spinamin E—**11**, echinamines A (**12**) and B (**13**)) of sea urchins *S. intermedius* and *S. mirabilis* [36].

**Table 2 marinedrugs-20-00611-t002:** Content of naphthoquinone pigments in the shells of sea urchins *S. intermedius* and *S. mirabilis* [36].

Naphthoquinone Pigments	Family Species	
	*Strongylocentrotus intermedius*	*Scaphechinus mirabilis*
	Content of Main Spinochromes, % of Pigment Sum	
** *Echinochrome A* **	-	89.1 ± 8.7
** *Spinochrome A* **	3.3 ± 0.3	-
** *Spinochrome B* **	4.0 ± 1.3	-
** *Spinochrome C* **	9.2 ± 1.5	-
** *Spinochrome D* **	16.5 ± 5.7	1.8 ± 0.9
** *Spinochrome E* **	3.1 ± 1.6	-

In addition, Ageenko and her colleagues developed an in vitro technology for inducing differentiation of pigment cells in culture (Table 3) [37]. The number of pigment cells was also evaluated during cultivation in different media (seawater—SW, in the coelomic fluid of intact sea urchins—CFn, in the coelomic fluid of wounded sea urchins—CFreg) and the qualitative composition of naphthoquinoid compounds in them (using MALDI and mass spectrometry with electrospray ionization). Spinochrome D and E were found in the pigment cells of the sea urchins *S. intermedius*, and cultured pigment cells of the sand dollar *S. mirabilis* contained echinochrome A and spinochrome E.

Polyhydroxy-1,4-naphthoquinones are characterized by a labile quinoid structure. This instability is caused by the presence of hydroxyl substituents of the naphthazarin cycle, which are subject to redox transformations. It is known that at physiological pH values, polyhydroxyl 1,4-naphthoquinones are one or divalent anions, and this can affect their reactions with ions and radicals [38].

The authors [39], using methods ultraviolet-visible spectroscopy and UPLC-DAD-MS, studied the stability of quinoid pigment compounds (ethylidene-6,6′-bis(2,3,7-trihydroxynaphthazarin, spinochrome dimer and spinochrome D) isolated from the sea urchin *S. droebachiensis*. With a change in the acidity of ethanol solutions of naphthoquinones and the time of their incubation, a change in the concentration of these pigments in the solution was also observed.

Thus, in order to create medicines based on naphthoquinones, there is a need for conditions for stabilizing the quinoid structure. This direction is very relevant in scientific research.

Shikov and his colleagues [30] described in detail the qualitative composition of naphthoquinoid pigments isolated from 33 different species of sea urchins (Table 4). The paper [40] presents the characteristics of the main quinoid pigment substances isolated from 19 species plants (Table 4). Sea urchins and plants contain a large supply of naphthoquinones, which exhibit different types of biological activity. It should be noted the variety of 1,4-naphthoquinoid pigments in sea urchins, which makes them more attractive for the creation of medicines and food additives.

### 2.3. The Main Ways of Biosynthesis of Quinoid Pigment Compounds

The biosynthesis of pigments of any class proceeds only along one main pathway with subsequent modification of the structure and the production of individual compounds. Quinoid compounds, unlike other pigments, are a biosynthetic heterogeneous group of substances. Thus, different organisms can synthesize the same pigment quinoid substance using different biosynthesis pathways [1].

Previously it was known that there were only three ways of naphthoquinone biosynthesis—polyketide, shikimate and mevalonate (isoprenoid). Quinones are oxidized derivatives of aromatic compounds and can be synthesized in a similar way.

Currently, it is known that in nature, naphthoquinoid pigments are formed as a result of other biosynthetic pathways [41,42].

#### 2.3.1. Polyketide Pathway of Quinone Biosynthesis

The polyketide pathway of quinone biosynthesis is a process of gradual chain elongation, similar to the biosynthesis of fatty acids (Figure 4) [42]. At the first stage of biosynthesis, condensation of acetyl-CoA and malonyl-CoA occurs. In addition, the elongation of the carbon chain is increased by the addition of C2 fragments of malonyl-CoA until the desired chain length is reached. Malonyl-CoA serves as an activated donor of acetyl groups. Thus, a polypeptide system is formed due to CO and CH_2_ groups with the elimination of a water molecule and the formation of an aromatic or quinoid molecule [42,43].

This is the pathway of biosynthesis of various quinones—benzoquinones, naphthoquinones, anthraquinones and higher quinones (fungal quinones, anthracycline, pyrromycinone from nine acetate and malonate fragments) [42,43,44,45,46,47,48,49,50].

#### 2.3.2. The Shikimate Pathway of Quinone Biosynthesis

The shikimate pathway is the main one in the biosynthesis of aromatic and colored quinones (naphthoquinones) and unpainted fragments of quinoid compounds in cells (plastoquinone, phylloquinone, menaquinone and ubiquinone) with the formation of shikimic acid [42,49,50].

The mechanism of quinone biosynthesis begins with the cyclization of 7-phospho-3-deoxy-D-arabinoheptulosonic acid, with the further formation of 5-dehydroquinic acid, which then turns into shikimic acid (A), and then into 5-phosphoshikimic acid. After that, phosphoenolpyruvate ether is attached to these acids and chorismic acid (B) is formed, which undergoes intramolecular rearrangement and turns into prephenic acid (C) (Figure 5) [42].

Thus, many natural quinoid pigment compounds formed due to precursors—shikimic, chorismic and prephenic acids (Figure 5) [42].

To date, the pathways of the biosynthesis of naphthoquinone pigments with the participation of shikimic acid have been studied in sufficient detail. The shikimate pathway of quinone biosynthesis is best studied by the example of 1,4-naphthoquinone biosynthesis [42,49,50].

#### 2.3.3. The Mevalonate Pathway of Quinone Biosynthesis

Substituted naphthoquinones and anthraquinones are formed from mevalonic acid and its derivatives by mevalonate biosynthesis [42,50]. A combination of two biosynthetic pathways (shikimate and mevalonate) is often used by plants (*Pyroleae* (*Dumort*, 1829)) to produce substituted naphthoquinones. The quinoid structure of 1,4-naphthoquinone – chimaphilin is derived from shikimate, and methyl substituents and benzoic acid atoms are derived from mevalonate (Figure 6) [42].

It should be noted that all quinoid compounds in plants and animals can be synthesized by various ways of biosynthesis [40]. For example, mushroom quinones are mainly formed using the polyketide pathway, while higher quinones often undergo a mixed biosynthesis pathway (shikimate and mevalonate pathways together).

#### 2.3.4. The Pathways of 1,4–Naphthoquinone Biosynthesis

The formation of 1,4–naphthoquinoid compounds in nature occurs through a cascade of complex reactions along several biosynthetic pathways.

In addition to the metabolic pathways described above, several other pathways of 1,4–naphthoquinone biosynthesis are known (Figure 7, Figure 8 and Figure 9): an acetate-polymalonate pathway, a 4HBA pathway (4–hydroxybenzoic acid pathway), a GPP pathway (geranyldiphosphate pathway), an HGA pathway (homogenizate pathway), an MVA pathway (mevalonic acid pathway), an OSB–pathway (*o*-succinylbenzoate pathway) and a phylloquinone pathway [40].

The biosynthesis of 1,4–naphthoquinones of plant and animal origin proceeds along the first (the acetate-polymalonate pathway) and the sixth pathway (the OSB pathway). The first six metabolic pathways provide the formation of quinoid compounds in the plant kingdom. The seventh pathway forms menaquinone in bacteria [40].

It should be noted that 1,4–naphthoquinoid compounds can be formed according to more complex joint biosynthesis schemes. For example, the first ring in the structure of 1,4–naphthaquinones is formed in one and two joint biosynthetic pathways—OSB; 4HBA/GPP; and HGA/MVA, from chorismate, a product of biosynthesis of shikimic acid. The difference between these three metabolic pathways of naphthoquinoid compounds lies in the precursors, from which the second already quinoid ring of 1,4–naphthoquinones is formed (Figure 7, Figure 8 and Figure 9) [40].

The most important pathway of biosynthesis of various 1,4-naphthoquinones is the OSB pathway, which consists of seven reactions leading to the formation of 1,4-dihydroxy-2-naphthoate (DHNA) from chorismate, a key precursor in metabolic synthesis [50,51,52,53,54,55,56,57,58,59]. It is metabolite that provides the formation of the first 1,4-naphthalenoid ring of the benzene part in all 1,4-naphthoquinone derivatives [60,61,62,63]. Further, under the influence of OSB-CoA ligase, the side succinyl chain turns into a CoA-ether, which, when exposed in DHNA-CoA synthase, cyclizes to form a second ring of the quinine part of the 1,4-naphthoquinone molecule [64,65,66].

The biosynthesis of quinoid pigments—spinochromes and echinochrome A—in echinoderms, as well as anthraquinones in crustaceans and insects, proceeds along an independent pathway, but more often along the polyketide pathway.

The study of the biosynthesis of echinochrome A was first carried out in the sea urchin *Arbacia pustulosa* (*Leske*, 1778) by French scientists [67], who determined the maximum inclusion of the radioactive label [2-^14^C] in the cyclic part of the echinochrome A molecule and the minimum inclusion in the side chain. Thus, it was suggested that the cyclization of the polyketide chain of five acetate groups occurs first, and then a side chain is formed. In addition, sea urchins contain polyketide synthases (multi-enzyme complexes) capable of synthesizing polyketide chains from acetic acid residues. It is possible that the biosynthesis of sea urchin spinochromes occurs with the participation of their own enzyme complex without the participation of endosymbionts [68].

At the same time, Ageenko and colleagues [69] showed the effect of shikimic acid on the expression level of genes for the biosynthesis of quinoid compounds—polyketide synthase (*pks*) and sulfotransferase (*sult*)—in embryos and larvae of the sea urchin *S. intermedius* at different stages of development and in some tissues of adult animals. Perhaps these data may indicate the passage of spinochrome biosynthesis in sea urchins along a combined pathway. The mechanism of biosynthesis of 1,4-naphthoquinone derivatives in animals is still poorly understood, unlike fungi, microorganisms and plants [1].

### 2.4. Biological Activity of Naphthoquinoid Pigments of Sea Urchins

Sea urchins, in particular *S. mirabilis* and *S. intermedius*, contain naphthoquinoid pigments. Echinoderm pigments and related compounds form a new class of highly effective phenolic-type antioxidants that exhibit high bactericidal, algicidal, antiallergic, hypotonic and psychotropic activity. Due to its unique antioxidant properties, echinochrome A has been of great interest to scientists around the world for more than 30 years [70,71,72].

#### 2.4.1. Anti-Oxidant Activity of Naphthoquinones

Echinoids are unique sources of various metabolites with a wide range of biological activity [73,74,75]. The biological functions of quinoid pigments are very diverse. First, quinones are involved in electron transfer [76,77,78]. Living organisms throughout their lives produce reactive oxygen species (ROS), which regulate all processes of vital activity. It is known that reactive oxygen species, depending on conditions, can affect cell division in various ways (stimulate or suppress), provoke cell differentiation or apoptosis, damage nucleic acids and proteins [79,80,81,82,83,84]. Thus, reactive oxygen species play an important role in the induction of free radical processes in the cell. Damage of DNA molecules by free radicals leads to violations of the nuclear apparatus and the appearance of somatic mutations [85]. Such violations of cells under the influence of the active form of oxygen can lead to their premature death and rapid aging of the body [86,87]. Currently, there are more than 100 diseases (atherosclerosis, diabetes, cancer, rheumatoid arthritis, hypertension, etc.) that have arisen as a result of the effects of free radicals on the cells of the body. It is known that the process of inhibiting the action of free radicals in the body is possible under the influence of various antioxidants (endogenous—enzymes, and exogenous—phenol compounds). Endogenous antioxidants include superoxide dismutase (*SOD*), catalase, glutathione-independent peroxidases and transferases that have removed organic peroxides [77,88]. Exogenous antioxidants—simple phenols, oxy-derivatives of aromatic compounds and naphthols—are effective interceptors of radicals [89]. Several thousand isolated phenol compounds exhibit pronounced antioxidant properties. Such a number of antioxidants include phenylalanine, tryptophan, vitamins E and K, and most animal and plant pigments (phenocarboxylic acids, flavonoids and carotenoids) [90]. In addition, phenolic antioxidants (α-tocopherol, bilirubin, lycopene and carotenes) are effective inhibitors of various forms of active oxygen (hydroxyl radical, singlet oxygen and superoxide anion radicals) [91]. Molecules of menaquinone (an electron transporter involved in anaerobic redox reactions generating ATP) and ubiquinone (an endogenous mitochondrial compound) are respiratory coenzymes that act similarly to vitamin E or vitamin K and are involved in electron transfer in microorganisms, plants and animals [92,93].

In conditions of oxidative stress, non-enzymatic low molecular weight antioxidants play an important role [77,92]. For example, fat-soluble α-tocopherol, located in the hydrophobic layer of biological membranes, inactivates fatty acid radicals.

It is known that a large number of biologically active substances that exhibit antioxidant properties are part of the gonads, tissues of internal organs, shell and needles of sea urchins. Their antioxidants can be activated by phospholipids of plasma membranes, chelate metals, intercept free radicals and inhibit lipoxygenase enzymes [93,94,95,96,97].

The most important pigment of sea urchins with high biological activity is echinochrome A. Moreover, other naphthoquinone pigments of sea urchins—spinochromes B, C, D and E—exhibit pronounced biological activity [98,99].

The study of quinoid pigments of sea urchins is widely carried out by scientists from Japan, China, Korea, Vietnam and Russia. Different types of sea urchins contain naphthoquinoid pigments of different composition, showing different biological activity. It is known that all spinochromes are capable of regenerative properties. It has previously been shown that spinochrome A and echinochrome A are inhibitors of hydroxylases (dopamine-β-hydroxylase and tyrosine hydroxylase), which are targets for the treatment of hypotension.

The antioxidant activity of spinochromes and echinochrome A was first studied by Russian scientists [100]. All quinoid pigments of sea urchins represent a new class of natural antioxidants. Isolated spinochromes A–E and echinochrome A from extracts of different species of sea urchins showed high antioxidant activity in models of inhibition of lipid substrate oxidation, chelation and reduction of iron ions, interception of hydrogen peroxide and superoxide radical anion, interaction with 2,2-diphenyl-1-picrylhydrazyl (DPPH) [19,20,101,102,103,104].

Besides, many quinones are synthesized by organisms in self-defense. For example, many insects secrete toxic and aggressive simple benzoquinones. Some fungi synthesize naphthoquinones, which exhibit antibacterial and antiviral properties. Substituted 1,4-naphthoquinone—marticin synthesized by the phytopathogenic fungus *Fusarium martii* (*Appel and Wollenw.,* 1910)—contributes to the destruction of the host plant. Some quinones are also capable of causing various allergic reactions in humans and other mammals.

#### 2.4.2. Anti-Bacterial Activity of Naphthoquinones

It is known that granules of coelomic fluid (red spherocytes) of some sea urchins [105] protect against microbial infection and are important when the shell is injured [106,107].

The anti-bacterial activity of some spinochromes of various species of sea urchins against marine bacteria (*Vibrio aesturianus* (*Pacini*, 1854), *Cobetia marina* (*Cobet et al*., 1970; *Arahal et al*., 2002), *Shewanella oneidensis* (*Venkateswaran et al*., 1999) and model bacteria (*Escherichia coli* (*Migula*, 1895), and *Bacillus subtilis* (*Ehrenberg*, 1835; *Cohn*, 1872) was studied. Echinochrome A and spinochromes showed high anti-bacterial activity against the studied bacteria [106]. Echinochrome A and spinochromes A–E have also been investigated for anti-microbial activity against other bacteria. It was found that this anti-bacterial activity is very high against the gram-positive bacteria of *Staphylococcus aureus* (*Rosenbach*, 1884) and fungal cultures of *Saccharomyces carlsbergensis* (*E.C. Hansen*, 1908), *Candida utilis* (*Berkhout*, 1923) and *Trichophyton mentagrophytes* ((*Robin*) *Blanchard*, 1853).

With the growth of the aquaculture industry, bacterial infections are increasing, which leads to the death of millions of juvenile organisms, and this explains the increased interest in understanding how marine organisms protect themselves. Previously, the influence of marine bacteria on the embryonic development of the sea urchin *S. intermedius*, larval survival, the number of pigment cells and the expression level of genes for the biosynthesis of pigment quinoid compounds *pks* and *sult* was studied [106]. As a result, it was shown that the incubation of embryos of the *S. intermedius* of various stages of development with 22 strains of bacteria of nine genera (*Aliivibrio* (*Beijerinck*, 1889), *Bizionia* (*Nedashkovskaya et al*., 2005), *Colwellia* (*Deming et al*., 1988), *Olleya* (*Mancuso Nichols et al*., 2005), *Paenibacillus* (*Paenibacillus Ash et al.*, 1994), *Photobacterium* (*Véron*, 1965), *Pseudoalteromonas* (*Gauthier et al*., 1995), *Shewanella* (*MacDonell and Colwell*, 1985) and *Vibrio* (*Pacini*, 1854)) lead to a slowdown in embryonic development, deformation and reduced survival of larvae. The expression of *pks* genes increased significantly after incubation of embryos with all these bacteria, and the expression of *sult* genes increased only after the incubation of sea urchin embryos with *Pseudoalteromonas* and *Shewanella* bacteria. In addition, the precursor of pigment biosynthesis, shikimic acid, also increased the resistance of embryos to the action of these bacteria. Thus, it was found that the specific genes *pks* and *sult*, as well as shikimic acid, are involved in the bacterial protection of sea urchins.

#### 2.4.3. Anti-Viral Activity of Naphthoquinones

The combined use of echinochrome A with ascorbic acid and α-tocopherol showed not only higher antioxidant, but also antiviral effects against tick-borne encephalitis virus (*TBEV*) and herpes simplex virus type 1 (*HSV-1*) than the use of echinochrome A alone [107,108].

#### 2.4.4. Anti-Inflammatory Activity of Naphthoquinones

In addition, the anti-inflammatory activity of spinochromes has been described in vitro [104]. Spinochromes A, B, E, a mixture of echinochrome A and spinochrome C increased the level of proinflammatory cytokine TNF-α in macrophages *J774* after stimulation with lipopolysaccharide. It should be noted that during in vivo experiments, echinochrome A also demonstrated a pronounced anti-inflammatory effect [109].

#### 2.4.5. Anti-Allergic Activity of Naphthoquinones

Currently, the antiallergic properties of quinoid pigment substances are still poorly studied. However, in the work [110] for the first time, the potential possibility of using an extract of pigments isolated from the shell of sea urchins *S. droebachiensis* as an antiallergic agent was shown. The author and colleagues have revealed a significant inhibitory effect of polyhydroxy-1,4-naphthoquinone extract of sea urchins *S. droebachiensis* with the allergic inflammation of the isolated ileum of a guinea pig.

It was also previously described the use of chrysophanol-8-O-β-D-glucopyranoside, a natural anthraquinone isolated from rhizomes of *Rheum undulatum* (*Polygonaceae*) (*Linnaeus*, 1753), as an antihistamine component in the treatment of asthma [111]. A similar property is possessed by the commercial drug “Disodium Cromoglycate” [111].

In addition, the antipruritic activity of 1,4-naphthoquinone derivatives is known [112,113].

#### 2.4.6. Cytotoxic Activity of Naphthoquinones

The cytotoxic activity of spinochromes has been investigated against human *HeLa* tumor cells. The results of the study showed low cytotoxicity of sea urchin naphthoquinoid pigments for these tumor cells [104].

#### 2.4.7. Study of Pharmacokinetic Properties of Naphthoquinones

An important stage in the study of various properties of biologically active substances is the study of the pharmacokinetics of these substances, i.e., their chemical transformations (absorption of BAS, their distribution, metabolism and excretion) in animals and humans.

Part of the review [114] is devoted to the description of the pharmacokinetic properties of echinochrome A isolated from sea urchins *S. mirabilis*.

The authors [115] investigated the absorption of the sodium salt of the naphthoquinone pigment echinochrome A (Histochrome) in rabbits after parabulbar and subconjunctival injections. The drug is quickly distributed over the eye without being absorbed into the blood.

The metabolism of echinochrome A in rats was also studied [116]. Rats were injected subcutaneously with a solution of echinochrome A for 10 days at 10 mg/kg, after which this pigment was converted into 2-methoxy-3,5,6,8-tetrahydroxy-7-ethyl-1,4-naphthoquinone and 3-methoxy-2,5,6,8-tetrahydroxy-7-ethyl-1,4-naphthoquinone.

Pharmacokinetic studies in humans have been conducted for a few compounds isolated from marine animals. The results of pharmacokinetic studies of echinochrome A are known when studying the effect of the human Histochrome drug [117]. This drug was administered intravenously (1%; 100 mg) to the subjects. As a result, a high volume distribution of the drug in plasma (5.7 l), low clearance (0.16 l/h) and an extended half-life (T1/2) up to 87.3 h were determined.

The study of the pharmacokinetic properties of biologically active naphthoquinones and other substances isolated from marine animals helps to investigate their metabolism and improve the bioavailability of poorly soluble compounds when creating drugs [118,119,120].

#### 2.4.8. The Prevention and Treatment of Cardiovascular Diseases, Disorders of Carbohydrate and Lipid Metabolism during Aging, Ophthalmological Substances

Cardiovascular diseases occupy the first place among human diseases. Currently, scientists around the world are actively developing drugs and biologically active additives for the treatment of patients with lipid metabolism disorders, which form the basis of cardiovascular pathology.

Modern therapy in medicine is aimed at normalizing carbohydrate metabolism, reducing the content and intensity of assimilation of saturated fats and cholesterol in food. An important direction is to reduce the production of free radicals, adhesion molecules, cytokines, reduce the formation of atherosclerotic plaques, as well as enhancing the antioxidant activity of the blood and human immunity. Currently, the most effective drugs for the prevention and treatment of coronary heart disease (CHD) are drugs of the statin group that have pleiotropic effects and block endogenous cholesterol synthesis by inhibiting 3-methyl-3-glutaryl-coenzyme-A-reductase.

In addition, statins reduce the intensity of oxidative stress, improve the functional activity of the endothelium of blood vessels, stabilize atherosclerotic plaques and reduce platelet adhesion. Despite this, for medical reasons, not all patients can take statins.

Echinochrome A, having the functions of antioxidants and vitamins of group K, can be part of medicines for the treatment of cardiovascular, oncological diseases; diabetes mellitus; liver pathology; aging; disorders of carbohydrate and lipid metabolism [73]. Echinochrome A and other polyhydroxyaphtoquinones isolated from the flat sea urchin *S. mirabilis* have anti-platelet properties, reduce cholesterol levels in the blood and the formation of atherosclerotic plaques, normalize erythrocyte membranes, reduce the accumulation of toxic peroxides in the cardiac muscle tissue, an" exhibit anti-viral and anti-microbial effects [103].

Based on the data obtained, new effective drugs with unique therapeutic properties Histochrome for cardiology and Histochrome for opthalmology were developed [121,122]. Histochrome interacts with free radicals and active oxygen forms in the first 10–20 min, stabilizing cell membranes [101,123,124,125,126].

The work of Shikov [30] describes the main effects of the drug Histochrome (solution of sodium salt of Echinochrome A) in the treatment of humans.

It is known that with the introduction of a single intravenous injection of Histochrome (100 mg), the aggregation of erythrocytes and platelets decreases, and there is an improvement in the structure of blood cells and stabilization of their function [127]. In addition, the use of the same concentration of Histochrome leads to inhibition of lipid peroxidation in plasma and a decrease in necrosis of heart tissues in acute phases of myocardial infarction [117,128]. With intramuscular administration of Histochrome (2 mL of 0.02% solution) for 10 days, patients with cardiovascular pathology experience modulation of the immune system [129].

Also, the use of Histochrome (0.5 mL, 0.02%, subconjunctival or peribulbar injection) leads to the epithelialization of defects in patients with corneal dystrophy and increased visual acuity (in 60% of cases), a decrease in the volume of edema [123], and to the complete resorption of retinal hemorrhages after traumatic intraocular hemorrhage [130,131].

#### 2.4.9. Naphthoquinones Are Analogues of Medicines

Echinochrome A, which has a similar structure to vitamins of groups K and C, exhibits vitamin properties [132].

In addition, echinochrome A exhibits a physiological effect similar to ascorbic acid [133], demonstrating a similar transport pathway into the cell. When echinochrome A penetrates the tissue, the oxygen content decreases and hydrogen peroxide is formed, which induces the release of transcription factors (PPARa, PPARß and PPARy) by peroxisomes, which reduce inflammation and regulate cellular metabolism.

#### 2.4.10. Using in Agriculture

When freezing embryonic sea urchin cells in liquid nitrogen, the use of echinochrome A in combination with exogenous lipids ensures high cell viability (75–80%) and the ability to synthesize pigment granules and spicules after thawing [132].

Echinochrome A is also used in agriculture for cryopreservation of sperm from farm animals [132].

#### 2.4.11. Biotechnological Using

Cultures of embryonic cells of sea urchins represent a new model system for obtaining cells and their directed differentiation in vitro. At the same time, it is possible to use artificial and natural substrates, unique biologically active substances from the tissues of marine organisms and various growth factors.

Currently, little is known about the genes of growth factors that are expressed in the tissues of marine invertebrates. For vertebrates, the key genes regulating the state of cells and ensuring a high level of proliferation of embryonic cells in culture are mainly two genes, *nanog* and *oct-4* [133,134]. One of these genes, *SpOct*, was previously found in the sea urchin *S. purpuratus*. In addition, we found the nanog gene in the genome of sea urchins, the expression of which was detected at the stage of the mesenchymal blastula [135]. The expression of foreign genes (yeast *gal4* gene and plant oncogenes *rol*) leads to abnormal development of sea urchin embryos [136,137]. The active proliferation of embryonic sea urchin cells in culture can be induced by the introduction of plasmids containing, for example, the *gal4* gene. After 2 months of cultivation, the naphthoquinoid pigment echinochrome A was detected in cells obtained from transfected embryos [137].

### 2.5. Food Supplements Based on Sea Urchins

#### 2.5.1. Food Supplements Based on Freeze-Dried Sea Urchin Caviar

Currently, food supplements are an important addition to human food, supporting a healthy lifestyle. In addition, they can be used in complex therapy for the treatment and prevention of various pathological diseases. Food supplements created on the basis of sea urchins are aimed at reducing the absorption of saturated fats and cholesterol by the human body with food. Such food supplements have a targeted effect on all stages of the atherosclerotic process [38].

For the prevention and treatment of patients with dyslipidemia, food supplements based on freeze-dried sea urchin caviar (BME) are of particular interest [38].

BME is a natural complex of biologically active substances containing polyunsaturated fatty acids (PUFA) (125 mg). The composition of this drug includes: omega–3 PUFA (more than 20%); fats (20%)—phospholipids (22–36%), lecithin (61–67%), triglycerides (60–75%); sialoglycolipids; carotenoids; micro- and macroelements (cobalt, copper, iron, iodine, etc.); vitamins (B1, B2, B12, C, K1, PP) [38].

To study the normalization of the blood lipid profile in dyslipidemia, Kovalev and his colleagues conducted a number of experiments using BME separately and together with different doses of atorvastatin. The experiment was carried out for 28 days on non-inbred white mice (males) (60 individuals of the same age, body weight 18–20 g; nursery “Stolbovoe”), which received an emulsion of cholesterol in vegetable oil (through a probe daily at the rate of 0.4 g/kg of animal body weight) and an atherogenic diet (lard (25%), butter (5%) of the weight of the daily diet, wheat porridge). The concentration of BME was 250 mg/kg (5 mg per animal). The study was carried out in accordance with all the requirements of the European Convention (European Convention for the protection of vertebrate animal used for experimental and other scientific purposes, 1986).

Experimental mice were divided into four groups: control (1), atherogenic diet (2), BME on the background of an atherogenic diet (3), BME after an atherogenic diet (4). As a result of the experiment, the content of high-density lipoproteins (HDL-C), triglycerides (TG), total cholesterol (HC), atherogenicity coefficient (CA), low and very low density lipoproteins (LDL-C, VLDL-C) were calculated [38].

In the blood serum of the group 2 mice relative to the group 1 animals, there was a significant increase in the amount of total HC, HC in atherogenic classes (LDL-C, VLDL-C, and TG) and CA. In the 3rd group of mice in their blood serum, there was a decrease in the levels of total cholesterol, LDL-C and VLDL-C, CA, with a simultaneous increase in the relative content of HDL. However, there was no decrease in the amount of total HC in the blood serum of the group 4 animals, but the levels of LDL-C and VLDL-C, TG and CA significantly decreased in comparison with similar indicators in the group 2 mice [38]. Thus, the use of BME as a dietary supplement has a more significant effect on the body against the background of an atherogenic diet than after it ends.

In addition, the dynamics of lipid metabolism indicators in patients with dyslipidemia (DLP) with the combined use of the drug atorvastatin (10 and 20 mg) with BME was studied. The therapeutic effect was achieved with the use of atorvastatin at a dose of 10 mg together with BME [38].

Moreover, the use of the therapeutic complex BME—atorvastatin (10 mg)—contributed to an increase in the level of HDL-C in the blood of patients, necessary to reduce the formation and stabilization of atherosclerotic plaques [38].

Thus, the combined use of BME with atorvastatin can enhance the pharmacological capabilities of statins and reduce the mortality rate from cardiovascular diseases.

#### 2.5.2. The Use of Food Supplements Based on Sea Urchin Caviar in the Treatment of Women during Menopause

In Russia, the biologically active supplement “Extra Youth” (“EY”) has appeared relatively recently (RU.77.99.11.003.E. 0011843.02.15) [38]. The composition of this drug includes sea urchin caviar, calcium alginate and rosehip fruits as a source of vitamin C. The drug is obtained by enzymatic hydrolysis technology.

The calcium salt of alginic acid is a polysaccharide of the marine brown algae *Laminaria japonica* (*Lamour*, 1813), consisting of two monomers—residues of polyuronic acids (D-mannuronic and L-guluronic) in different proportions. Alginic acid is a source of dietary fiber and calcium, and it improves the digestive process; binds; and removes heavy metals, radionuclides, toxins, allergens from the body.

Rosehip has a variety of vitamins and mineral salts, its fruits contain flavonoids (hyperoside, kaempferol, quercetin, quercitrin, lycopene, rutin). Vitamin C in the composition of this drug enhances the anti-inflammatory and effect and shows an antioxidant effect. Moreover, thanks to the composition of this dietary supplement, it has a positive effect on the musculoskeletal system, is a proflactic agent for inflammatory diseases of cartilage tissue and osteoporosis. In addition, “Extra Youth” also shows cosmetic effects and promotes flexibility of hair and nails.

Due to the properties of all the components of this drug, its study was conducted in the complex therapy of women with hormonal disorders during menopause.

The design of the study was that menopausal women took the drug “EY” for 30 days. Further, to assess the degree of changes, the patients answered questions from a specially developed test system. As a result, in patients after taking the drug “EY”, it was determined: 1. in 66%, insomnia disappeared; 2. in 64%, nervousness and a feeling of depression and despondency disappeared; 3. in 75%, a feeling of constant fatigue disappeared; 4. in 65%, night sweating significantly decreased; 5. in 66%, the skin condition improved; 6. in 55%, joint pain decreased; 7. 63% have significantly rarer hot flashes. At the same time, changes in clinical blood and urine tests were not detected, and clinically significant changes in the level of sex hormones in the blood were also not detected. Thus, the results of the paraclinical examination showed that when taking the drug “EY”, significant positive changes occurred in the state of women’s health.

## 3. Materials and Methods

We conducted the writing of this review using various literary sources, including world databases. Literature search was conducted in *Web of Science*, *Scopus*, *Pubmed*, *Scientific Electronic Library* (Russian database—elibrary.ru). All literary sources are listed in the “References” section including publications from 1885 to the present.

## 4. Conclusions

Naphthoquinoid pigments of sea urchins are a promising source for the production of drugs with various pharmacological activities. Echinochrome A is used for the prevention and treatment of cardiovascular diseases, disorders of carbohydrate and lipid metabolism during aging. Echinochrome A is also used in agriculture. Sea urchin pigments could become the basis for the development of new natural drugs. There are great prospects for the future development of naphthoquinone pigments of sea urchins into drugs with rich pharmacological activities and the limitations for further research connect with sea urchin catch limits or with *in vitro* researches.

## Figures and Tables

**Figure 1 marinedrugs-20-00611-f001:**
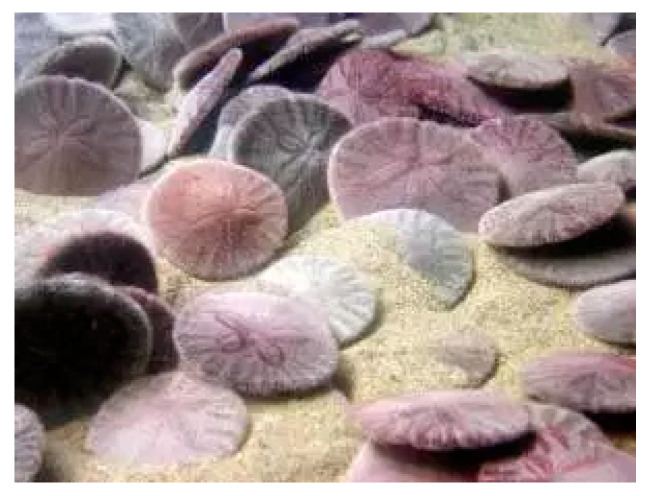
*Scaphechinus mirabilis* (Agassiz, 1863).

**Figure 2 marinedrugs-20-00611-f002:**
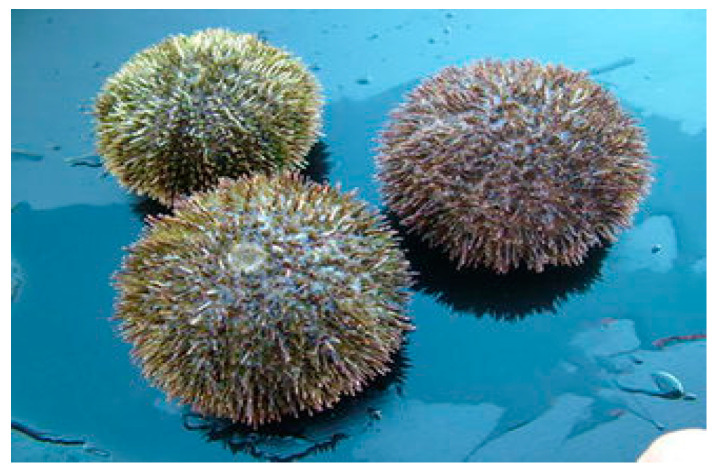
*Strongylocentrotus intermedius* (Agassiz, 1863).

**Figure 4 marinedrugs-20-00611-f004:**
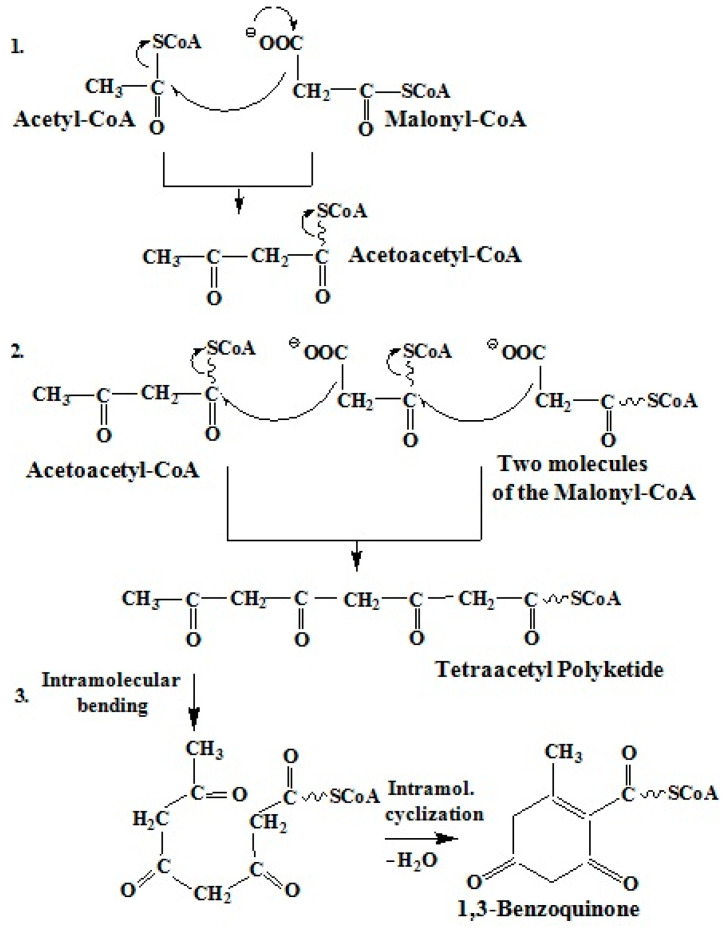
Diagram of the polyketide pathway of 1,3-benzoquinone biosynthesis [42].

**Figure 5 marinedrugs-20-00611-f005:**
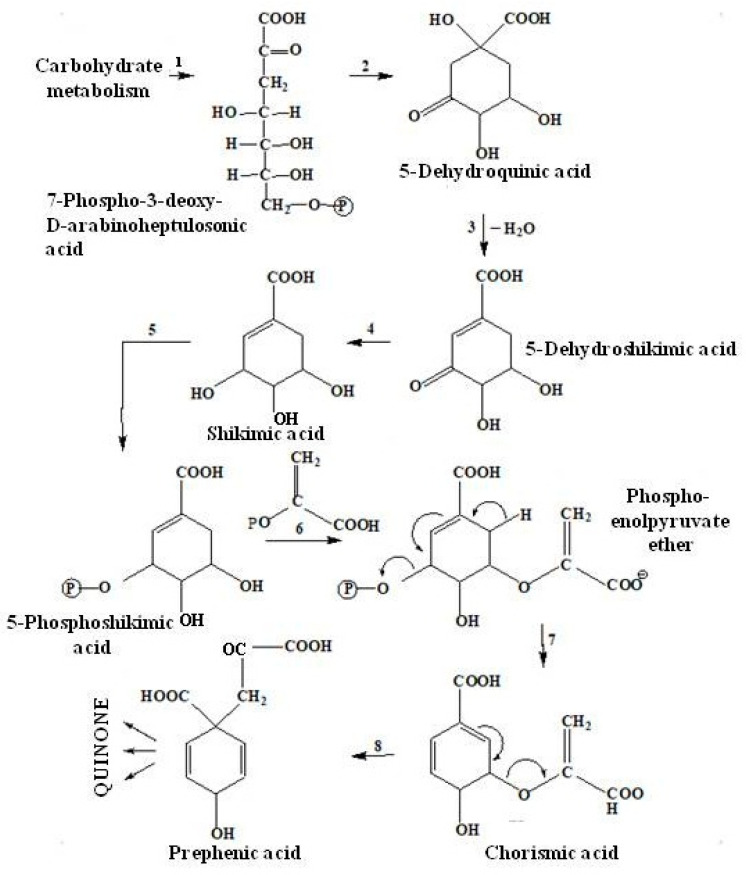
Diagram of the shikimate pathway of quinone biosynthesis [42].

**Figure 6 marinedrugs-20-00611-f006:**
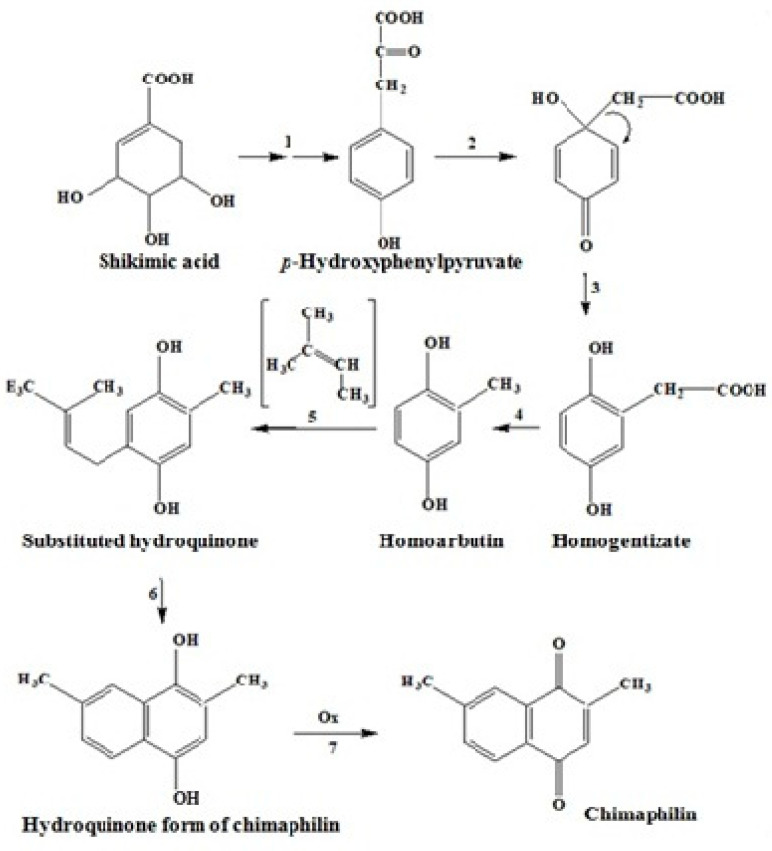
Diagram of the mevalonate pathway of chimaphilin biosynthesis [42].

**Figure 7 marinedrugs-20-00611-f007:**
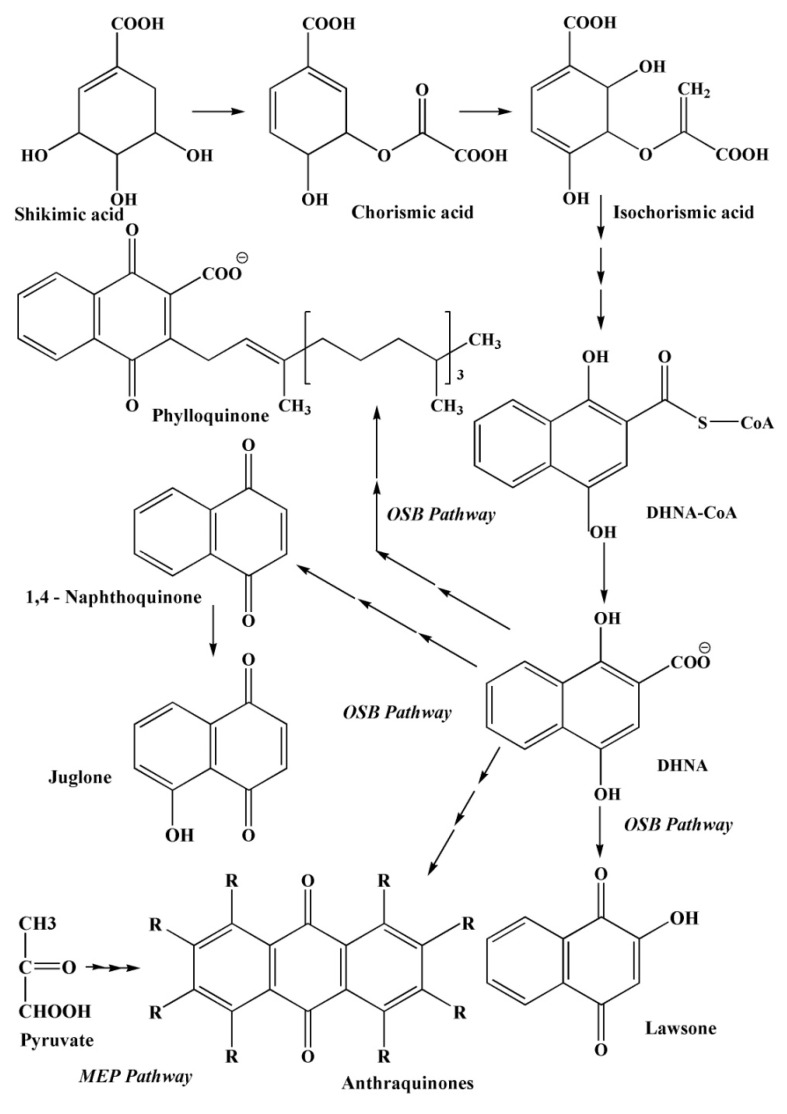
Diagrams of the OSB and MEP pathways of quinone biosynthesis [40].

**Figure 8 marinedrugs-20-00611-f008:**
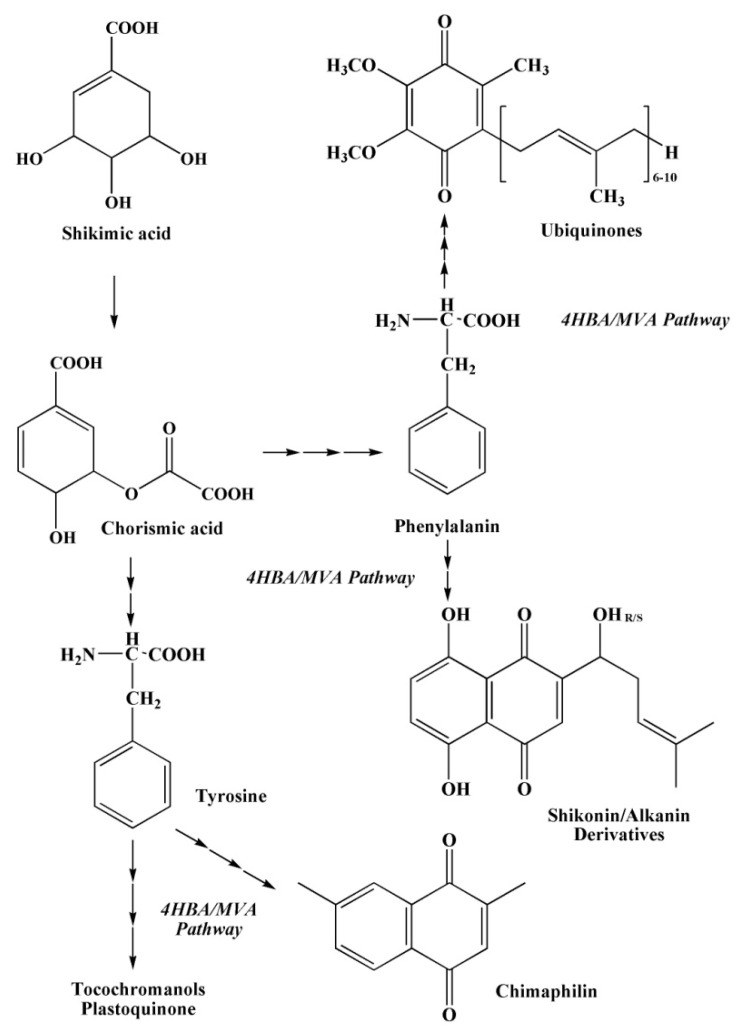
Diagrams of the 4HBA/MVA pathways of quinone biosynthesis [40].

**Figure 9 marinedrugs-20-00611-f009:**
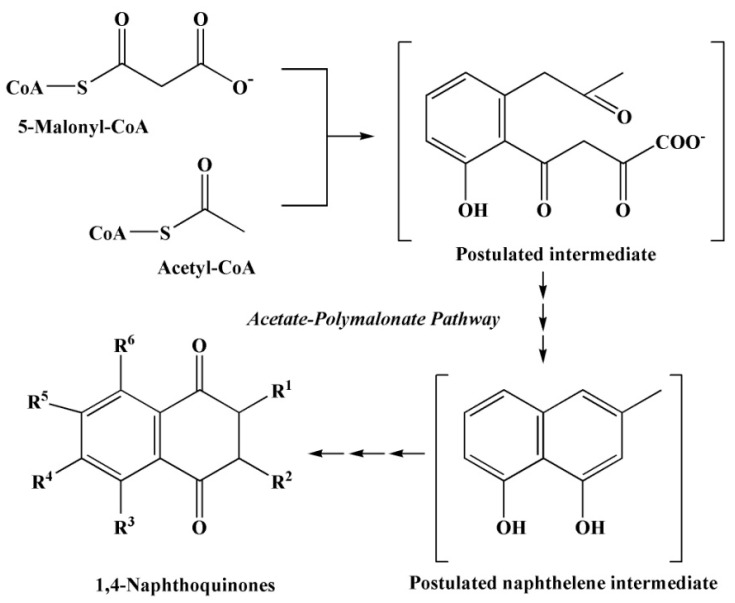
Diagrams of the acetate-polymalonate pathway of quinone biosynthesis [40].

**Table 1 marinedrugs-20-00611-t001:** The main bioactive pigments of sea urchins *S. mirabilis* and *S. intermedius*.

№	Name of Naphthoquinone Pigment	Structural Formula
**1**	Echinochrome A	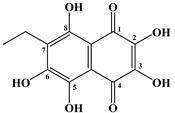
**2**	Spinochrome A	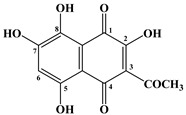
**3**	Spinochrome B	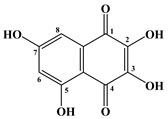
**4**	Spinochrome C	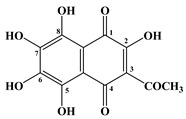
**5**	Spinochrome D	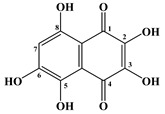
**6**	Spinochrome E	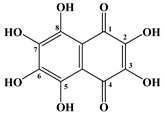

**Table 3 marinedrugs-20-00611-t003:** Content of naphthoquinone pigments in cultivated cells of sea urchins *S. intermedius* and *S. mirabilis* in different media: SW—seawater; CFn—coelomic fluid obtained from intact sea urchins; CFreg—coelomic fluid obtained from injured sea urchins. Data are presented as the mean ± standard error from two independent experiments (ESI MS) [37].

Naphthoquinone Pigments	Cell Culture Medium		
	SW	CFn	CFreg
	Fresh Biomass of Cells (mg/g)		
	*Strongylocentrotus intermedius*		
*Spinochrome E*	0.024 ± 0.004	0.013 ± 0.002	0.015 ± 0.002
*Spinochrome D*	0.052 ± 0.006	0.044 ± 0.005	0.062 ± 0.007
	** *Scaphechinus mirabilis* **		
*Spinochrome E*	0.021 ± 0.003	0.062 ± 0.007	0.054 ± 0.006
*Echinochrome A*	0.250 ± 0.026	0.640± 0.061	0.540 ± 0.055

**Table 4 marinedrugs-20-00611-t004:** Composition of naphthoquinone pigments of sea urchins and plants.

№	Naphthoquinone Pigments of Sea Urchin	Naphthoquinone Pigments of Terrestrial Plants
1	Echinochrome A	Lawsone
2	Spinochrome A (Spinochrome B3;Spinochrome M; Spinochrome M1; Spinochrome Aka2; Spinochrome P)	Lapachol
3	Spinochrome B(Spinochrome B1; Spinochrome M2;Spinochrome N; Spinochrome P1)	Alizarin
4	Spinochrome C(Spinochrome B2; Spinochrome F;Spinochrome F1; Spinochrome M3; Spinone A; Isoechinochrome)	Shikonins
5	Spinochrome D (Spinochrome Aka; Spinochrome Aka1)	Alkannins
6	Spinochrome E	Chimaphilins
7	Spinochrome G	Plumbagin
8	Spinochrome S	2-methoxy-1,4-naphthoquinone
9	2-Hydroxy-3-acetylnaphthazarin	Droserone
10	2,3,7-trihydroxy-6-ethyljuglone	Juglone
11	2-hydroxy-6-ethyljuglone	7-Methyljuglone
12	Naphthopurpurin	Anthrasesamones
13	6-Ethyl-2-hydroxynaphthazarin	
14	6-Acetyl-2,7-dihydroxyjuglone	
15	6-Acetyl-2-hydroxynaphthazarin	
16	Mompain	
17	Ethylmompain	
18	6-Ethyl-3,7-dihydroxy-2-methoxynaphthazarin	
19	6-Ethyl-2,7-dihydroxy-3-methoxynaphthazarin	
20	3-Acetyl-2,7-dihydroxy-6-methylnaphthazarin	
21	Echinamine A	
22	Echinamine B	
23	Aminopentahydroxynaphthoquinone	
24	Spinamine E	
25	Spinazarin	
26	Ethylspinazarin	
27	Tetrahydroxydimethoxynaphthoquinone	
28	Namakochrome	
29	Ethylidene-6,6′-bis(2,3,7-trihydroxynaphthazarin)	
30	Ethylidene-3,3′-bis(2,6,7-trihydroxynaphthazarin).	
31	Anhydroethylidene-6,6′,-bis(2,3,7-trihydroxynaphthazarin)	
32	Anhydroethylidene-3,3′-bis(2,6,7-trihydroxynaphthazarin)	
33	Mirabiquinone A	
34	Pyranonaphthazarin	
35	Acetylaminotrihydroxynaphthoquinone	
36	Spinochrome dimer	
37	Spinochrome B sulfate derivative	
38	Spinochrome E sulfate derivative	
39	Spinochrome A—Iso 2	
40	Spinochrome D—Iso 1	
41	Spinochrome D—Iso 3

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
