# Peer review of "Quinoid Pigments of Sea Urchins Scaphechinus mirabilis and Strongylocentrotus intermedius: Biological Activity and Potential Applications"

_marinedrugs, 2022, doi:10.3390/md20100611_

Round 1

Reviewer 1 Report

I have read the manuscript and I have some questions and suggestions.

1. Please prepare a section "Materials and Methods" indicating: years of literature search, keywords, databases.

2. In the headings of figures 2, 4, 5 and 6, indicate the literary reference.

3. For the "Creation of biologically active additives based on sea urchins" section, use the correct term - food supplements.

4. According to the topic of the manuscript, namely the biological activity and application of quinoid pigments from the sea urchins Scaphechinus mirabilis and Strongylocentrotus intermedius, please summarize the data in the form of a table/tables indicating the quinoid pigment, test system, model, dose/concentration range, positive /negative control, results in numbers, literature reference. Discuss the difference between pigments from Scaphechinus mirabilis and Strongylocentrotus intermedius and pigments isolated from various other sea urchins.

5. In section 2.3.2, provide a diagram of the shikimate pathway of quinone biosynthesis, which is specific for pigments from Scaphechinus mirabilis and Strongylocentrotus intermedius.

6. Discuss the scheme shown in Figure 4 or exclude it.

7. For section 2.3.3, give the scheme of mevalonate pathway of chimaphilin biosynthesis, which is specific for pigments from Scaphechinus mirabilis and Strongylocentrotus intermedius.

8. Section 2.4.3 does not contain information on drug production. Please correct.

9. The terms "metabolic correctors", "Vitamin-like compounds of naphthoquinones" are not correct. Use accepted terms.

10. To increase interest in the topic of quinoid pigments, please include an additional section on clinical use, indicating results, reference drugs, etc.

11. Quinoid pigments are very complex molecules in terms of their stability and pharmacokinetics. Discuss the stability of quinoid pigments from Scaphechinus mirabilis and Strongylocentrotus intermedius with pigments from other sources (e.g. https://doi.org/10.1080/14786419.2017.1290617).

12. Assess pigment bioavailability and pharmacokinetic data (eg, https://doi.org/10.3390/md18110557). Compare data. Show the prospects for the use of quinoid pigments from Scaphechinus mirabilis and Strongylocentrotus intermedius in comparison with pigments from other sea urchin species.

13. Compare pigments from Scaphechinus mirabilis and Strongylocentrotus intermedius with quinoid pigments from other marine organisms/terrestrial plants. Specify the advantages of quinoid pigments in sea urchins.

Author Response

Dear Editors,

We are grateful to the reviewers of Marine Drugs for detailed and friendly revision of our manuscript, and for valuable comments. We agree with all comments and improved the manuscript accordingly. We hope that the revised manuscript became clearer and more easily followed by readers.

Below we list the reviewer’s comments and explain how we changed the manuscript.

Reviewer 2 Report

Comments and Suggestions for Authors

The manuscript entitled “Quinoid Pigments of Sea Urchins Scaphechinus mirabilis and Strongylocentrotus intermedius: Biological Activity and Potential Applications” by Ageenko and co-workers described the main ways of biosynthesis of quinoid pixel compounds, the biological activity of naphtoquinoid pixels of sea urchins, and the various pharmacological activities of naphthoquinone compounds and their applications in other fields. However, the article would need significant restructuring in order to be publishable, as reading it can be quite confusing. Therefore, I do not think that this manuscript is suitable for publication in its current form. I have outlined my most prominent concerns as below.

There is a point that I am very confused about in this review. What is the relationship between naphtoquinoid segments, echinochrome A, and echinochrome? Is it a subordinate relationship or other troubles? The author needs to explain.

In part 2.4.1, the authors mentioned that quinoid pigments have been studied by scientists in many countries and have shown different biological activities. Is there any specific experiment to prove this argument, and can some specific experiments be added to confirm it?

The authors need to add some comments or the prospects for the future development of naphthoquinone pigments of sea urchins into drugs with rich pharmacological activities and the limitations for further research.  

In addition, the following are some details and grammatical problems that will confuse others when reading the article.

1、       In line 23 of page 1, "well know" should be "well-known".

2、       In line 40 of page 1, please delete "and" between transfer and etc.

3、       In line 54 of page 2, "echinocrome" should be "echinocromes".

4、       In line 78 of page 2, please add “are” between larvae and carried.

5、       In line 85 of page 2, "sea grasses" should be "seagrasses".

6、       Figure 1 should be in the middle of the article.

7、       Table 1 should be changed to three line table and beautified.

8、       Figures 2, 4, and 5 should be showed in chemical structures.

9、       Figure 3 should be in the middle of the article and beautified.

10、     In line 202 of page 6, "a ascetate-polymalonate" should be "an ascetate-polymalonate".

11、     In line 204 of page 6, "a HGA" should be "an HGA", "a MVA" should be "an MVA".

12、     In line 204 of page 6, "a OSB" should be "an OSB".

13、     In line 221 of page 6, please add “of” between influence and OSB-CoA.

14、     In line 222 of page 6, please add “to” between exposed and DHNA-CoA.

15、     In line 223 of page 6, "from" should be "form".

16、     In line 359 of page 9, please add a space between period and A.

17、     In line 370 of page 9, "of" should be "around".

18、     In line 396 of page 10, please delete a space between compond and s.

19、     In line 402 of page 10, please delete "into" between penetrates and the.

20、     In line 421 of page 10, please add “the” between of and mesenchymal.

21、     In line 478 of page 11, "activity" should be "activities".

Overall, I do think that topic is very interesting and I wish the authors the best of luck on thei further endeavors.

Author Response

(The authors gave the same response as above.)

Reviewer 3 Report

Dear author

This manuscript needs tor following revisions:

1. The English editing and grammar

2. the scientific names must be italic form with trinomial names, genus+species+authors.

3. The images of the urchins needed to better the manuscript.

4. The meta analysis and systematic review needed to accepteble this manuscript.

Author Response

(The authors gave the same response as above.)

Round 2

Reviewer 1 Report

The authors made the necessary corrections. I believe that the manuscript in a present form can be published in the journal Marine Drugs.

Reviewer 2 Report

The authors have revised the manuscript accordingly.

Reviewer 3 Report

Dear author

Goodluck